# Cancer and Diabetes: Predictive Factors in Patients with Metabolic Syndrome

**DOI:** 10.3390/diagnostics13162647

**Published:** 2023-08-11

**Authors:** Mihai Cosmin Stan, Daniel Georgescu, Ciprian Camil Mireștean, Florinel Bădulescu

**Affiliations:** 1Medical Oncology Department, Vâlcea Emergency County Hospital, 240156 Râmnicu Vâlcea, Romania; 2Medical Oncology Department, University of Medicine and Pharmacy of Craiova, 200349 Craiova, Romania; mc3313@yahoo.com (C.C.M.); fbadulescu2001@yahoo.com (F.B.); 3Department of Informatics and Medical Statistics, University of Medicine and Pharmacy of Craiova, 200349 Craiova, Romania; d_g_home@yahoo.com; 4Railways Clinical Hospital, 700506 Iasi, Romania

**Keywords:** diabetes, cancer, anemia, albumin

## Abstract

Background and Objectives: A growing number of epidemiological studies have suggested that diabetes mellitus may increase cancer risk and is implicated in numerous other metabolic and inflammatory disorders. The increase in proinflammatory cytokines plays a major role in insulin resistance and leads to hypoalbuminemia and micro- and macrovascular diabetes complications, including kidney disease and anemia. This study aimed to investigate the utility of carcinoembryonic antigen (CEA), C-reactive protein (CRP), serum albumin level, hemoglobin, and lactate dehydrogenase (LDH) as biomarkers for cancer risk, and the biological implications of diabetes on the evolution and prognosis of oncological patients. Material and Methods: We conducted a retrospective, longitudinal, observational study on a total group of 434 patients, of which 217 were diagnosed with a form of cancer and type two diabetes as a comorbidity, and the other 217 were a control group without diabetes. These patients were admitted to the oncology clinic. In subgroups, the same number of patients was considered, depending on the location of the oncological pathology. Anemia, hypoalbuminemia, elevated lactate dehydrogenase, glycated hemoglobin, and C-reactive protein levels are more pronounced in subjects with type two diabetes and cancer. Conclusions: The presence of diabetes negatively affects the clinical and biological prognosis of cancer patients.

## 1. Introduction

A growing number of epidemiological studies have found that diabetes mellitus increases the risk of developing many types of cancer [1]. The association between these two diseases was first discussed over 75 years ago. Recent and extensive evidence has shown that diabetes is associated with an increased risk of cancer and a higher mortality rate in cancer patients [2]. An epidemiological study that enrolled 420 patients with type two diabetes mellitus concluded that the most frequent association between diabetes and neoplasia was observed for lung, breast, pancreatic, and colorectal cancers [3]. Lactate dehydrogenase (LDH) levels, complete blood count, carcinoembryonic antigen (CEA), glycated hemoglobin (HbA1c), and C-reactive protein (CRP) are accessible biomarkers, forming part of the standard patient assessment; they could be predictive factors for cancer in diabetic patients and can guide the oncologist in terms of patient prognosis at the onset of treatment [4]. We believe that these biomarkers can also be used by a diabetes doctor to be able to suspect the presence of cancer, especially when an unexplained glycemic imbalance occurs. Developing colorectal cancer is 27% more possible in patients with type two diabetes mellitus than in non-diabetic patients [5]. Various studies have also demonstrated that the risk of colon cancer recurrence is higher in diabetic patients [5].

One study of 47 patients with diabetes and colorectal cancer showed an average HbA1c value of 6.0%, while 45% of them had an HbA1c of at least 7% [5]. There is insufficient information on CEA in colorectal cancer patients with diabetes. Several studies showed a statistically significant correlation between elevated CEA levels and diabetes, as well as a correlation between serum CEA levels and HbA1c levels [5]. Insulin resistance and relative insulin deficiency are characteristic of type two diabetic patients; they are often overweight and elderly with diabetes oncoming, compared with type one diabetic subjects. Obesity promotes insulin resistance and is considered one of the main reasons for the current diabetes epidemic [5]. Albumin is a major protein synthesized in the liver. Energy intake is a very important factor in the normal physiology of albumin production. In fact, reduced serum albumin levels are observed in diseases associated with malnutrition, while high serum albumin levels are associated with metabolic syndrome, an indicator of obesity and overeating. In addition, a link between serum albumin and insulin resistance has recently been suggested [6].

Long-term type two diabetes mellitus is associated with about 1.5 times increased risk of pancreatic cancer [7]. A causal relationship between diabetes and pancreatic cancer is also supported by prediagnosis measurements of glucose and insulin levels in prospective studies [7]. Insulin resistance and associated hyperglycemia, hyperinsulinemia, and inflammation are the underlying mechanisms contributing to the development of diabetes-associated pancreatic cancer. The mechanism of the relationship between diabetes and pancreatic cancer is elusive and could include metabolic, hormonal, and immunologic changes that influence tumor growth. Insulin resistance and compensatory hyperinsulinemia are maybe the most suspected mechanisms underlying the relationship between type two diabetes mellitus and pancreatic cancer [7]. 

Diabetes was associated with a significantly more pronounced risk of lung cancer compared with patients without diabetes when the analysis was restricted to studies that accounted for smoking status. In contrast, this association disappeared when the analysis was restricted to studies that did not consider smoking status [8]. Subjects with type one and type two diabetes had a higher fasting plasma lactate level than those without diabetes [9]. 

Chronic hyperlactatemia maintained by increased lactate formation from adipocytes in obese individuals has been found to precede the onset of diabetes [10] and might contribute to the appearance of the disease. Taken together, these data suggest that chronic hyperlactatemia may indicate the early stages of insulin resistance. Several epidemiological analyses have shown that high lactate levels may predict the onset of diabetes.

Lactic dehydrogenase prognostic value in patients with lung cancer was investigated by some researchers, but the findings were inconclusive. According to some studies, patients with lung cancer have worse prognoses when their LDH levels are higher. Some researchers discovered that this correlation was insignificant [11]. Higher pretreatment LDH levels have been associated with poorer overall survival in subjects with lung cancer [11].

Anemia is the most common hematologic change in patients with malignancies [12]. It may be the first diagnostic clue for underlying malignant disease and could contribute to symptomatology and treatment decisions. Cytokine production associated with the tumor is an important factor in the appearance of anemia in cancer patients. A malignant tumor can also affect bone marrow fibrosis, which can also result in anemia. Bone marrow is known for its rich blood supply and is, therefore, a common site for metastasis to develop [13]. Bone marrow fibrosis, originally characterized by Wartofsky and Burman, was originally called “low T3 syndrome” [13]. Breast malignancy, prostate, and lung are the types of cancer most often involved, although almost all types of cancer can lead to this complication. Normocytic-normochromic anemia occurs frequently in patients with a variety of inflammatory disorders, with many contributing factors.

Several studies have suggested the importance of inflammatory biomarkers in the onset of diabetes and the monitoring of antidiabetic treatment, without necessarily being related to a form of cancer. Thus, elevated C-reactive protein levels have been associated with an increased risk of the development of diabetes [14], and LDH is a biomarker of glycemic variability in diabetes monitoring [15]. There was also a significant increase in CEA values in patients with unbalanced diabetes, with average glycosylated hemoglobin values above 9% [16].

The objectives of the study were to assess the utility of CEA, CRP, serum albumin level, hemoglobin, and LDH as biomarkers of cancer risk and the biological implications of diabetes on the evolution and prognosis of oncological patients.

## 2. Materials and Methods

We performed a retrospective, longitudinal, observational study over a period of 2 years between 2016 and 2017, on two equal groups of patients, with 217 already having type two diabetes and being recently diagnosed with breast, lung, colorectal, or pancreatic cancer (the most frequent types of tumors associated with type two diabetes), and the other 217 without diabetes. All of them were hospitalized in the Oncology Medical Clinic of the Clinical County Emergency Hospital Craiova, Romania.

Criteria for inclusion: patients with a maximum performance status of two, hospitalized in the Oncology Medical Clinic of Emergency Clinical County Hospital Craiova (Romania) at the initiation of oncologic treatment, with the diagnosis of lung, breast, colon, or pancreatic cancer, who signed an informed consent. These patients were divided into two subgroups: subgroup 1 with type two diabetes, and subgroup 2 without diabetes.

Criteria for exclusion: both patients with other types of tumors, and patients with other types of inflammatory, acute, or chronic conditions, were excluded; also, patients for whom insufficient data were recorded or who did not sign an informed consent.

The study group was represented by 217 patients, already with type two diabetes, who were hospitalized in the Oncology Medical Clinic of the Emergency Clinical County Hospital Craiova (ECCHC), in the period of 2016–2017, with a performance status of a maximum of two, to begin oncologic specific treatment: 81 patients with lung cancer, 45 patients with colon cancer, 38 patients with pancreatic cancer, and 53 patients with breast cancer. After that, another equal group was formed, composed of 217 patients without type two diabetes. All data, for all 434 patients, were collected from the medical records of patients hospitalized in the Oncology Medical Clinic of ECCHC.

In this study, at the first admission, we evaluated the values of hemoglobin, CRP, HbA1c, LDH, alkaline phosphatase (FAL), serum albumin, and CEA in subgroup 1, with diabetes, and we compared them with the values found in subgroup 2, without diabetes. The HbA1c values in subgroup 2 were less than 5.7%, so type two diabetes and prediabetes were excluded [5,7], not only through HbA1c, but also through personal history.

The following Emergency Clinical County Hospital of Craiova analyzers were used in the study: COULTER DXH for hemoglobin, MINDRAY BS 800 for CRP and LDH levels, ARCHITECT C8000 for serum albumin, and COBAS E411-2 for CEA and HbA1c.

Statistical analysis was performed using Microsoft Office Pack - Microsoft Excel 2000 to collect data and obtain graphics and average values, and the MATLAB v.9.0 (2016) Student T-Test program to test the normality of data and expose the statistically significant differences between the two analyzed groups. The average values in the study group are compared with those of patients in the group without diabetes, a *p*-value < 0.05 being considered statistically significant. A confidence coefficient was also used, with an ‘r’ value close to 1 being considered a positive correlation.

Ethical concerns: the protocol of the study was approved by the Ethics Committee of the University of Medicine and Pharmacy of Craiova, approval number 39/20.01.2023.

## 3. Results

### 3.1. General Results

The average values in the study group, compared with those of patients in the group without diabetes, were strongly correlated, and the *p*-value result was 0.002.

In subgroup 1, the 217 patients with type two diabetes and cancer, the average value of hemoglobin was 11.3 g/dL at diagnosis of cancer, vs. 12.4 g/dL for the group without diabetes (*p* = 0.0001) (Figure 1 and Figure 2). The normal range considered was 12.5–14.5 g/dL.

Lactic dehydrogenase studied in the entire group was more abundant in the diabetes subgroup at cancer diagnosis (average 337 U/L) versus the group without diabetes (292 U/L) (*p* = 0.0033) (Figure 3 and Figure 4).

Glycated hemoglobin (HbA1c) values in subgroup 1 had rates between 5.7% and 13.1%, with an average value of 8.57%, which suggests poor diabetes control in relation to neoplasia. Non-diabetic patients were considered those with HbA1c values lower than 5.7%.

Many patients in the studied group had an average albumin level of 28.9 g/L, showing the nutritional status of patients with diabetes and cancer. In the group without diabetes, the average value of albumin was 35.3 g/L, showing that metabolic disorders were already present (*p* < 0.001) (Figure 5 and Figure 6).

A positive correlation in the entire analyzed group was identified between LDH values and serum albumin values (r = 0.76), but also between LDH and hemoglobin values (r = 0.82), especially in those patients diagnosed in stage IV (Figure 7 and Figure 8). Also, a positive correlation between hemoglobin and albumin levels was found (r = 0.99, Figure 9).

C-reactive protein levels in subgroup 1 were significantly higher when compared with the group without diabetes, with an average value of 73.47 mg/L, and 38.3 mg/L in subgroup 2, results which suggest a higher inflammation level in patients who had as comorbidity type two diabetes (*p* < 0.001) (Figure 10 and Figure 11).

Also, a positive correlation between hemoglobin and C-reactive protein was found (Figure 12).

Also, correlations between LDH and CRP levels (r = 0.83), and LDH and FAL (r = 0.72), were found. All these correlations suggest a sustained inflammatory response of the body in the presence of type two diabetes, more pronounced as the body is more hematologically and nutritionally balanced.

Average CEA levels for pancreatic cancer were higher in the diabetic group (39.23 ng/mL vs. 12.95 in the control group); in the colon cancer group, diabetic patients had a significantly lower rate of CEA, compared with the control group (44.63 ng/mL vs. 288.85 ng/mL—Figure 13).

### 3.2. Study of Biological Tests by Tumor Localization

#### 3.2.1. Lung Cancer 

In 81 patients with diabetes, the average value of hemoglobin at lung cancer diagnosis was 11.5 g/dL, versus 12.5 g/dL for the group without diabetes, and patients without diabetes (*p* = 0.003). Lactic dehydrogenase levels were not significantly increased in comparison with the group without diabetes (308 U/L in diabetic patients vs. 330 U/L in non-diabetic subjects). The average albumin levels for diabetic patients with lung cancer were 28.5 g/L and in the non-diabetic group they were 34.9 g/L (*p* < 0.001). C-reactive protein levels were also significantly elevated in patients from subgroup 1 (average value 91 mg/L vs. 40.2 mg/L in the group without diabetes) (*p* = 0.0022). The glycated hemoglobin level was between 5.7% and 13.1%, with an average value of 12%.

#### 3.2.2. Colon Cancer 

In subgroup 1, the average value of hemoglobin was 11.2 g/dL, compared with the non-diabetic group’s average of 12.1 g/dL (*p* = 0.01). Lactic dehydrogenase levels were elevated in the study group when cancer was diagnosed (average value 411 mg/dL) compared with the group without diabetes (273 mg/dL) (*p* = 0.0017).

A major difference was observed for albumin levels, showing that the nutritional balance for colon cancer in association with type two diabetes is fragile (average albumin 28.3 g/L in diabetic patients vs. 36.6 g/L in the non-diabetic group) (*p* = 0.00023). C-reactive protein levels were also increased in the study group vs. the group without diabetes (59.9 mg/L vs. 39.2 mg/L, *p* = 0.008). Carcinoembryonic antigen levels, studied in pancreatic and colon cancer patients, were significantly elevated in non-diabetic colon cancer cases, with an average value of 288 ng/mL vs. 44.3 ng/mL in diabetic patients.

#### 3.2.3. Pancreatic Cancer

Anemia at diagnosis was more common in pancreatic cancer cases with type two diabetes as comorbidity, with an average value of hemoglobin of 11.3 g/dL, against 12.4 g/dL for non-diabetic pancreatic cancer cases (*p* = 0.006). The average LDH level was 344 mg/dL in the diabetes subgroup, and 305 mg/dL in the non-diabetes subgroup, indicating a worse prognosis of the disease in diabetic patients.

The carcinoembryonic antigen value was 3-fold higher in diabetics (average value 39.2 mg/dL), against 12.9 mg/dL in non-diabetic patients. Inflammatory markers were higher in diabetes patients (CRP average value 63.3 mg/dL vs. 38.7 mg/dL in the non-diabetes group).

There was no significant difference in albumin values (29.6 g/L in the study group vs. 34.1 g/L in the group without diabetes, *p* = 0.00091).

The average HbA1c value in diabetes patients at diagnosis of pancreatic cancer was 8.69%.

#### 3.2.4. Breast Cancer

In the breast cancer subgroup, the average value of hemoglobin was lower in the group of diabetes patients (11.05 g/dL and 12.6 g/dL in non-diabetic patients) (*p* = 0.00011). Lactic dehydrogenase levels were elevated in the diabetes group, 312 mg/dL, versus 223 mg/dL (*p* = 0.00029), which suggests an increased inflammation in diabetes patients, confirmed by CRP values (64.9 mg/dL vs. 31.4 mg/dL in non-diabetes group, *p* = 0.0022). 

Denutrition was more pronounced in patients in the diabetes group (29.6 g/L vs. 35.1 g/L for non-diabetes, *p* < 0.0001). The average value of HbA1c was 8.49%.

Treatment and survival: In the analyzed group of oncological patients with diabetes, only 41 of them performed at least 3 months of specific oncological treatment, which included surgery, chemotherapy, or external irradiation. In comparison, in the control group, 88 patients could be treated for at least 3 months.

The median duration of oncological treatment in cancer patients with associated diabetes was 4 months, while in the control group, it was 6 months. The median duration of survival in the diabetic patient group was 7 months, while the median survival in the control group was 10 months (Figure 14).

## 4. Discussion

This study was designed to assess the utility of CEA, CRP, serum albumin level, hemoglobin, and LDH as biomarkers of cancer risk (colon, lung, breast, and pancreatic cancer) in obese and/or diabetic patients, and to study correlations between these markers and the prognosis of cancer patients with associated type two diabetes at the onset of treatment.

In our study, it was found that anemia, hypoalbuminemia, elevated lactate dehydrogenase, glycated hemoglobin, and C-reactive protein levels are more pronounced in subjects with type two diabetes and cancer (*p* = 0.002). These biomarkers may be an indicator of a patient’s inflammatory state for diabetic patients, and a neoplasm can be sought.

Hematologic markers that reflect systemic inflammation can also be used to predict the prognosis of cancer. Systemic inflammatory responses play a significant role in carcinogenesis, cancer progression influencing tumoral responses under oncologic treatment, and survival [4].

Recent studies show that type two diabetes is an independent risk factor for the progression of several cancers. Although these two diseases have an important number of common risk factors, the connection between them is still not well understood, posing a challenge for clinical management [9].

The incidence of diabetes has been increasing in the past two decades (464.237 million in 2019) and is expected to increase further, as it is estimated to rise to more than 700 million by 2045 [17,18].

Given the increasing interest of the European community and the Romanian government in implementing comprehensive prevention programs against the spread of cancer and type two diabetes [19], and the fact that many diabetic patients develop a malignancy during their lifetime [4], we also wanted to investigate the impact of type two diabetes on the oncological patient, and how predictive factors can be used. Diabetes is a concomitant disease that can also influence the therapeutic response to cancer therapy [4]. The duration of treatment in a patient who also has diabetes was shorter compared to those patients without diabetes, which also has an impact on survival, with its median being 7 months in the diabetic group, compared to 10 months in cancer patients without diabetes.

Lactic dehydrogenase levels in the diabetic group for each type of cancer was studied, but the most significant difference was in the colorectal cancer (411 mg/dL vs. 273 mg/dL in the group without diabetes) and breast cancer subgroups (312 mg/dL vs. 223 mg/dL in non-diabetic cases). Increased lactate alters the microenvironment, provides nutrients to cancer cells, and leads to acidosis, inflammation, angiogenesis, immunosuppression, and radiation resistance [20]. Bonuccelli et al. (2010) showed that ketones and lactate promote tumor growth and metastatic disorders, which may explain why diabetic subjects have increased cancer incidence and poor prognosis due to increased lactate production [20]. The body of a human contains LDH in a variety of tissues. Usually, anaerobic environments like the intratumoral environment are where the reaction between lactic acid and pyruvic acid takes place. The environment around tumors contains high levels of LDH, reflecting anaerobic glycolytic metabolism. The patients in the high-metastatic-score group had significantly higher serum LDH concentrations than those in the low-metastatic-score group, according to other studies [20].

A major reason for the development of cancer is that the immune system loses its ability to effectively eliminate aberrant cells. High levels of lactate have a deleterious effect on immune cells infiltrating the tumor [21]. Finally, lactate is an inflammatory mediator [22] and could be a biomarker of inflammatory processes promoting tumor development [23]. This specific inflammatory microenvironment also promotes tumor metastasis [4], leading to a negative prognosis [24].

Malignancies can also attack the bone marrow (bone marrow fibrosis), which can also lead to anemia. The bone marrow has a rich blood supply and is, therefore, a frequent site for the development of metastases [13]. Breast cancer, prostate, and lung malignancies are most frequently associated with anemia, although anemia may be encountered in almost all cancers. Normocytic-normochromic anemia is also common in patients with several inflammatory diseases. Iron may be plentiful in the bone marrow but is not absorbed and does not enter the bloodstream, making it unavailable for erythropoiesis [12]. Due to diabetes mellitus, nephropathy may occur, which further undermines renal production of erythropoietin, contributing to anemia [25]. Anemia in diabetic patients affects the quality of life and is associated with disease progression and the development of comorbidities [26].

The increase in proinflammatory cytokines plays a major role in insulin resistance and leads to the occurrence of micro- and macrovascular diabetic complications. Increasing IL -6 results in an anti-erythropoietic effect, as this cytokine alters the sensitivity of precursor cells to erythropoietin and promotes cellular death of immature erythrocytes, leading to a further decrease in the number of circulating erythrocytes and a decrease in circulating hemoglobin [12]. Our study highlighted that higher CRP values were found in diabetic cancer patients (average value 73.47 g/L) compared with non-diabetic patients (average CRP value 38.3 g/L). Positive correlations between inflammatory biomarkers, serum albumin, and hemoglobin values were found, suggesting a more sustained inflammatory response, as the nutritional and biologic status is more balanced.

Genetic syndromes, inflammatory bowel diseases, a history of abdominal radiation therapy, dietary factors (red meat, alcohol, high-fat/low-fiber diet), and tobacco use are the most incriminated factors for developing digestive tube malignancies [27,28]. There are risk factors (obesity and physical inactivity) common to those with diabetes mellitus. Inflammation associated with diabetes may also contribute to the development and progression of colorectal cancer [29]. CRP levels were more significant in the diabetic group than in the non-diabetic group with colorectal cancer (49.9 mg/L vs. 39.1 mg/dL), supporting this fact. In a previous study, the researchers reported that increased lactate dehydrogenase release in HT-29 colon cells has been linked to the biological mechanisms of diclofenac-induced cell death, while chrysin alleviated this effect [30,31].

There was no difference in survival between patients with colorectal cancer who had diabetes and those who did not, according to a recent case-control study on the relationship between type two diabetes and colorectal cancer [26]. The same study discovered that diabetic patients with colorectal cancer had a higher CEA than the group without diabetes and that their value decreased much more quickly under specific treatments than the group of patients without type two diabetes. This finding may have an impact on treatment choices [26].

For pancreatic cancer, there is increasing evidence that inflammation plays an important role in its development [32]. Our study found that inflammatory markers are higher in diabetic patients (CRP average value 59.96 mg/dL vs. 39.2 mg/dL in the non-diabetic group). Inflammatory pathways are frequently activated by obesity and macronutrient intake. Glucose and fat intake can trigger inflammation by increasing oxidative stress and activating transcription factors such as nuclear factor-κB, activating protein-1, and early growth response-1 [29]. Some adipocytokines are key compounds involved in innate immunity, inflammation, apoptosis, and metabolism. Significant levels of proinflammatory cytokines promote angiogenesis, tumor progression, and metastasis [7]. A high level of glycemia is directly related to the development of an inflammatory state, as evidenced by the increased expression of proinflammatory cytokines such as IL -6, TNF-α, and NFkB [29]. Studies show that the longer the disease persists and/or the poorer the glycemic control, the more severe the inflammatory process. The increase in proinflammatory cytokines plays a major role in insulin resistance and leads to the occurrence of diabetic macrovascular, cardiovascular, and microvascular complications and anemia [25,26].

Denutrition is a known major problem both for diabetics and cancer patients. This study revealed that low plasmatic albumin levels were found in diabetic patients (average value of albumin 28.9 g/L), compared with non-diabetic patients (35.3 g/L).

Uncontrolled type two diabetes in cancer patients was also specific in our study, confirmed by an average value of HbA1c of 8.57%.

Several studies suggest an important association between the depth of anemia and HbA1c levels in patients with diabetes, supporting the hypothesis that anemia is more common in poorly controlled diabetics [12,33]. In our study, hemoglobin and HbA1c levels were not correlated.

Cell proliferation and tissue damage are induced by chronic inflammation [34]. Diabetes and cancer are in a vicious cycle with each other, and lactate plays a central role in this interaction. Insulin resistance/diabetes and cancer lead to high lactate levels; conversely, high lactate levels promote the development and progression of diabetes and cancer [9].

The study limitations are represented by the limited biological tests available, with few options of state-settled biomarkers for the basic biological assessment of the hospitalized patient which can be used for this purpose.

## 5. Conclusions

This study was not conducted by clinical trials, and its results from current practice, which show a statistically significant correlation between inflammatory biomarkers in the presence of cancer and type two diabetes, can be applied in clinical practice and can lead to improved oncological patient prognosis. Hemoglobin, LDH, CRP, and albumin levels could be used as predictive factors for cancer patients with associated type two diabetes, highlighting the importance and impact of metabolic and inflammatory disorders encountered in these chronic diseases. Anemia was strongly correlated with type two diabetes and lung, colon, breast, and pancreatic cancers; there was a higher incidence of anemia among poorly controlled diabetics. Lactate dehydrogenase levels in the diabetic patient group were significantly elevated in type two diabetes associated with cancer, versus the non-diabetic patient group. Inflammation biomarkers, hemoglobin values, and serum albumin levels are also strongly correlated, suggesting a more sustained inflammatory response, as the nutritional and biologic status is more balanced. Better diabetes control can contribute to a favorable prognosis of malignancies. The presence of diabetes negatively influences the evolution and prognosis of cancer patients, median treatment period, and survival rate.

## Figures and Tables

**Figure 1 diagnostics-13-02647-f001:**
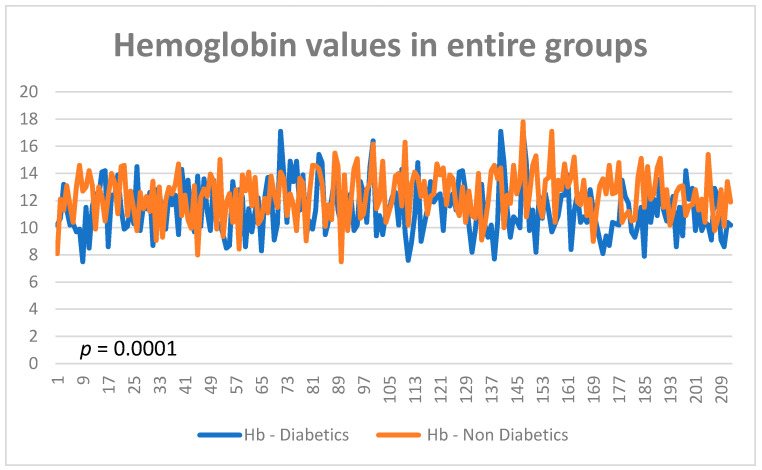
Hemoglobin values in entire groups.

**Figure 2 diagnostics-13-02647-f002:**
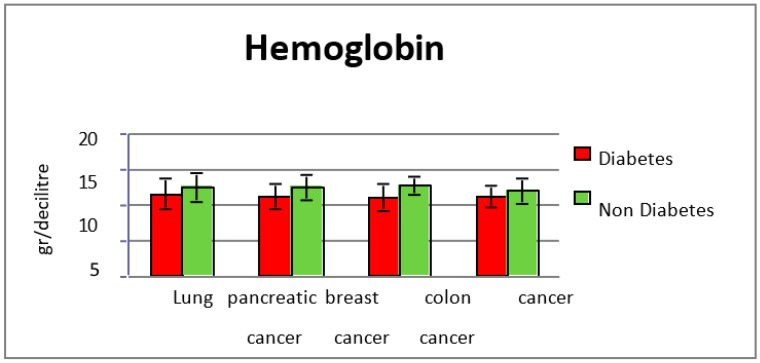
Hemoglobin values in the study groups.

**Figure 3 diagnostics-13-02647-f003:**
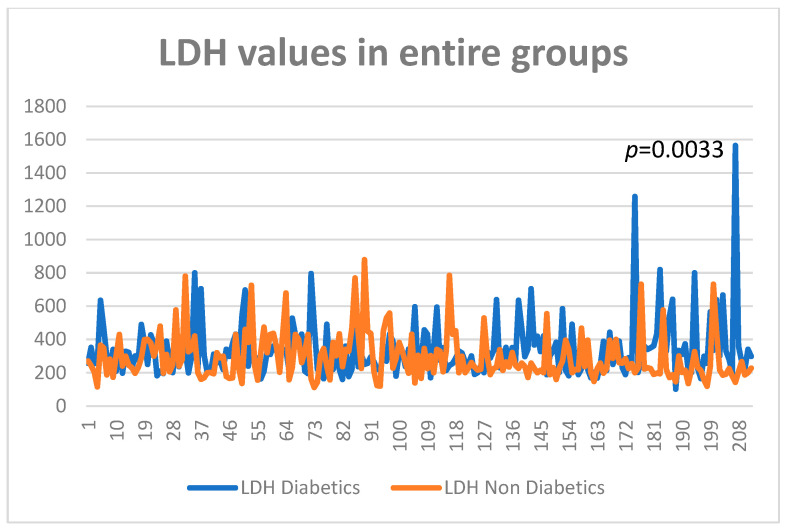
LDH values in entire study groups.

**Figure 4 diagnostics-13-02647-f004:**
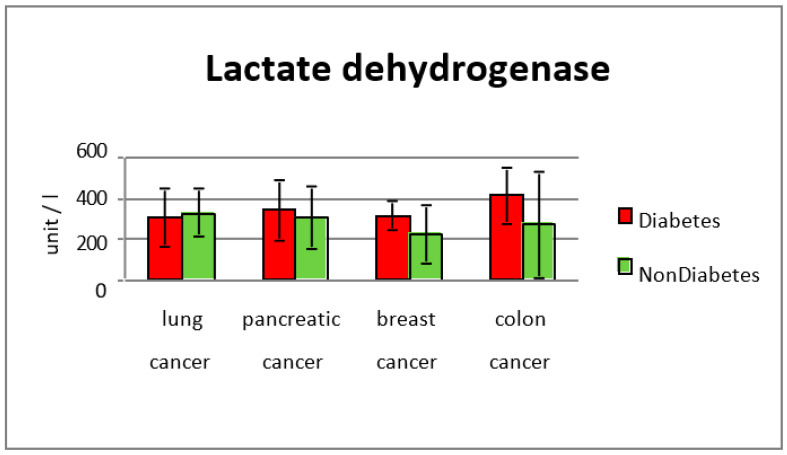
LDH values in the study groups.

**Figure 5 diagnostics-13-02647-f005:**
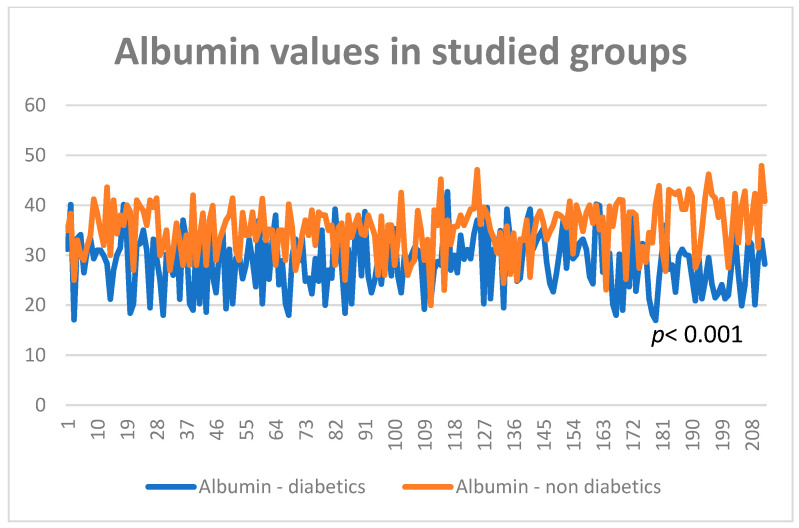
Albumin values in entire study groups.

**Figure 6 diagnostics-13-02647-f006:**
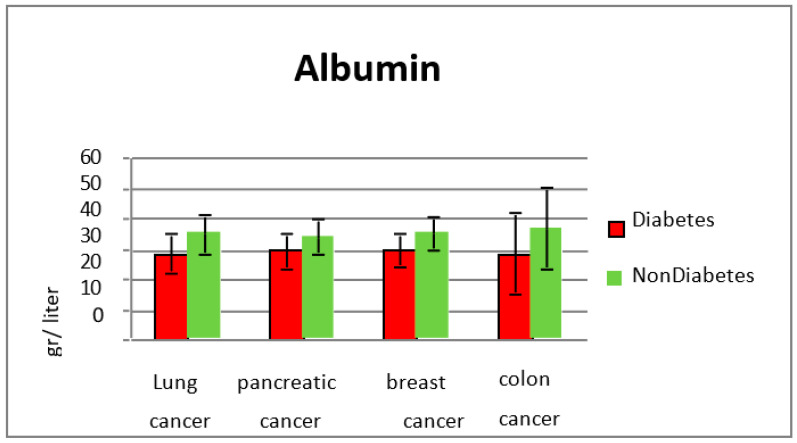
Albumin values in the study groups.

**Figure 7 diagnostics-13-02647-f007:**
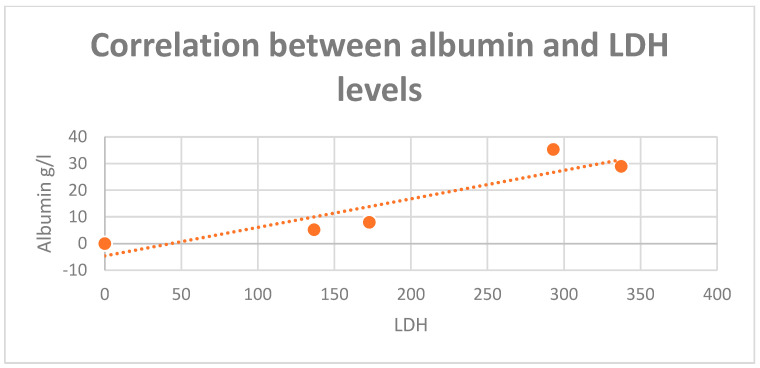
Positive correlation between albumin and LDH levels (r = 0.91).

**Figure 8 diagnostics-13-02647-f008:**
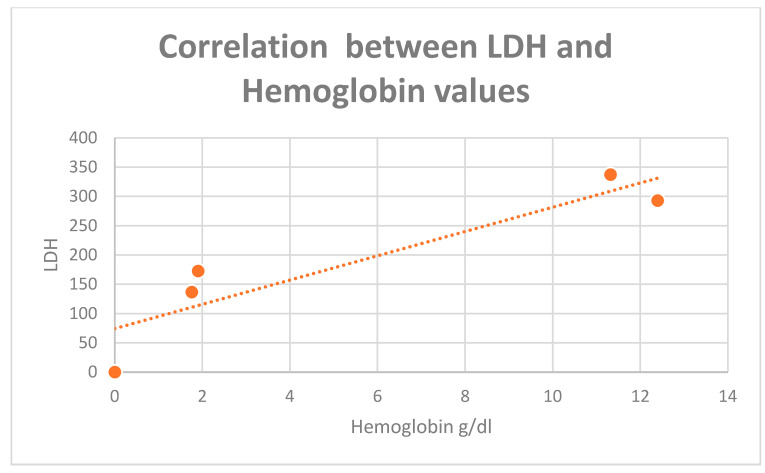
Positive correlation between LDH and hemoglobin values (r = 0.91).

**Figure 9 diagnostics-13-02647-f009:**
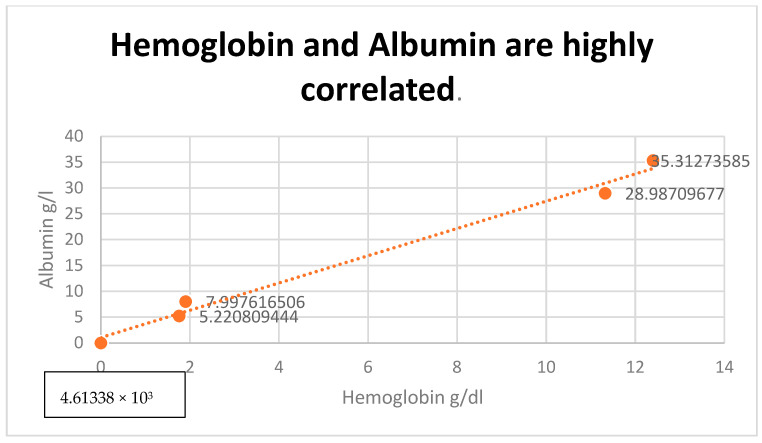
Positive correlation between hemoglobin and albumin levels (r = 0.99).

**Figure 10 diagnostics-13-02647-f010:**
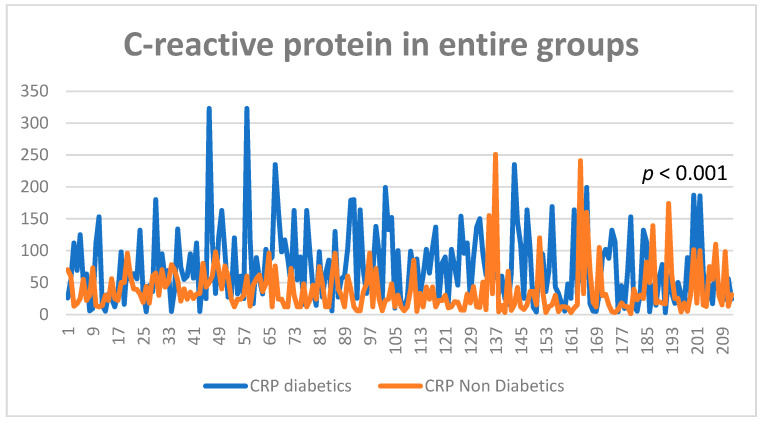
CRP values in entire study groups.

**Figure 11 diagnostics-13-02647-f011:**
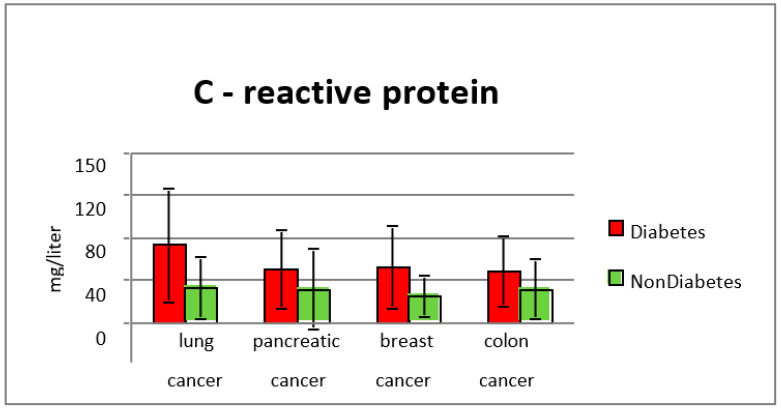
CRP values in the study groups.

**Figure 12 diagnostics-13-02647-f012:**
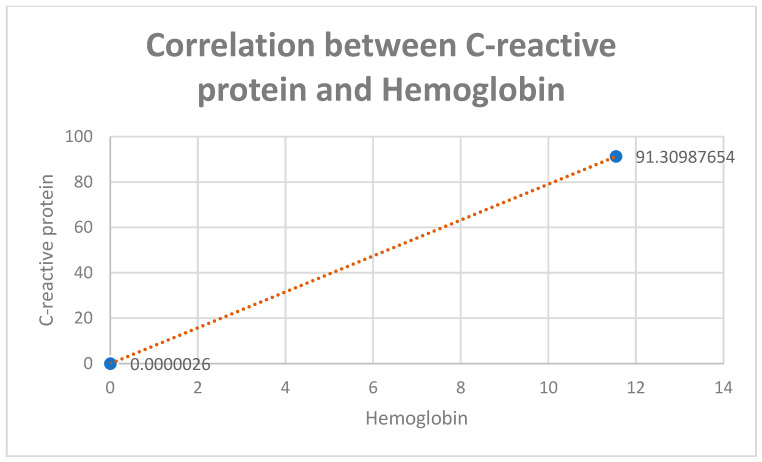
CRP values in the study groups correlated with hemoglobin values.

**Figure 13 diagnostics-13-02647-f013:**
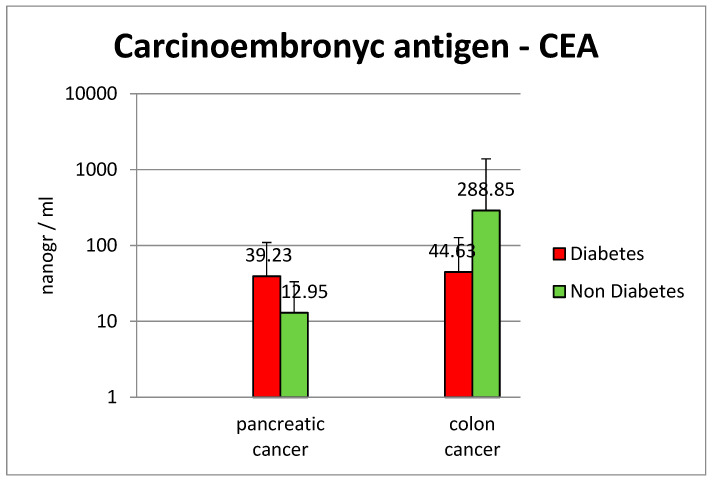
Mean values for carcinoembryonic antigen—CEA.

**Figure 14 diagnostics-13-02647-f014:**
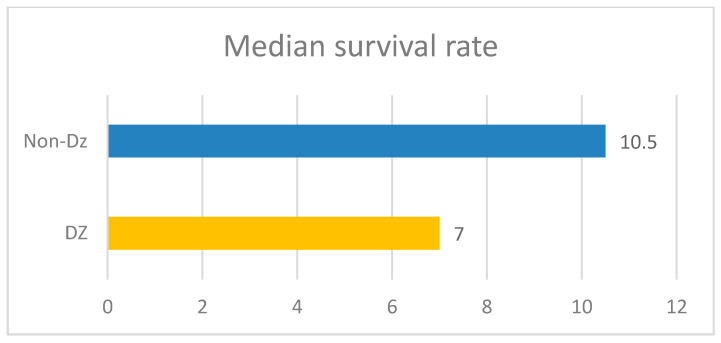
The median survival rate in diabetic patients versus the control group.

## Data Availability

Not applicable.

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
