# Peer review of "Cancer and Diabetes: Predictive Factors in Patients with Metabolic Syndrome"

_diagnostics, 2023, doi:10.3390/diagnostics13162647_

Round 1

Reviewer 1 Report (New Reviewer)

The authors present an interesting perspective about how some metabolic markers may present an opportunity for the early diagnose of cancer in patients with diabetes. The work as merit and its theme is very contemporary and adequately framed within the scope of the Journal. It should appeal for researchers and medics working in oncology, general medical care, as well as scientists working in the nutrition and metabolism areas. The data collection methodology is well explained and seems adequate, bibliography is up-to-date, and this paper could represent a valuable resource to researchers conducting science in these fields. There are some issues with the paper that should be addressed before publication, and changes that could make the paper substantially better.

Main comments:

I have two main concerns with the paper.

First and foremost, is the absence of a group of only diabetic patients. This would be another essential to fully appraise results and could be very informative to see how biomarkers/condition of the patients evolves from only diabetes to diabetes and cancer. In the absence of the possibility to produce such experimental group, a new section show be produced, I guess in the introduction, describing the available picture regarding the studied biomarkers/parameters (CEA, CRP, serum albumin level, hemoglobin, and LDH) in the context of diabetes. Are there basal values available for diabetes (only) patients for these markers? How do they compare with healthy individuals? I should this issue should be addressed and would greatly improve the scientific quality of the paper.

Another important issue is the absence of references in many sentences/claims in the discussion. The authors should make an effort to sustain all these claims in the discussion, in order to make their paper a much better resource for readers and researchers interested in finding information on the subject.

Finally, I just want to mention that the quality of the English could be improved (maybe a revision by a native speaker?). I have included some changes/suggestions regarding these issues in the minor comments.

Minor comments:

Line 13: “A growing number of epidemiologic studies have found that…”. Replace with ” A growing number of epidemiologic studies have suggested that…

Line 17: “…to investigate the utility of CEA, CRP, serum 17 albumin level, hemoglobin, and LDH as biomarkers…”. Do not use abbreviations here, before establishing what they stand for.

Line 32: “Recent and multiple evidence had shown…”. Replace with “Recent and multiple evidence has shown…”

Line 35: “the most frequent association between diabetes and neoplasia was for lung, breast…”. Include “observed”: “the most frequent association between diabetes and neoplasia was observed for lung, breast…”.

Line 82: “LDH’s prognostic value…” Do not start sentences with abbreviations, use full names. Check the rest of the document, has I have found this issue in several different occasions.

Line 131: “There were used the following ECCHC laboratory analyzers:”. Replace with “The following ECCHC analysers were used in the study:”. Also, please define what ECCHC means.

Line 134: “Statistical analysis was realized using…”. Replace with: Statistical analysis was performed using…”

Line 145: “The average values in the study group, compared with those of patients in the group without diabetes, were strongly correlated, p-value resulted was 0.002.” Average values of what? Please specify.

Line 147: “In sub-group 1 of patients with type 2 diabetes and cancer, compared with sub-group 2, the average value of hemoglobin was 11.3 g/dL at the moment of diagnosis of cancer, vs 12.4 g/dL for the group without diabetes (p = 0.0001) (Figure 1).” This is not actually in Figure 1. You can include another set of columns for total values, and then point to figure 1. The same for following figures 2 and 3.

Line 150, Figure 1: Please include statistical significancy (the usual asteriscs, if you will) in the figures (columns). That will allow the figures to be appraised more directly, independently from the text. The same for figures 2, 3 and 7.

Line 160: “…was more pronounced in the diabetes subgroup…”. Use abundant, instead: “…was more abundant in the diabetes subgroup…”.

Line 181, Figure 6: Figure headline (phrase at the top of figure) should be redacted according to the previous figures for consistency purposes.

Line 191, Figure 8: Please change according to the previous correlation graphics: points and a tendency line, again for consistency within the paper.

Line 242: “In the analyzed group of oncological patients with diabetes, only 41 of them performed at least 3 months of specific oncological treatment, which included surgery, chemotherapy, or external irradiation. Instead, in the control group, 88 patients could be treated for at least 3 months.”, until the end of the Results section: If this part is not related with breast cancer only and are general observations from the study (as they seem), a new section should be created (this should not be included in section 3.2.4).

Line 272: “…estimated at more than 700 million in 2045.” Replace with: “…estimated to become more than 700 million by 2045.”

Line 286: “Increased lactate alters the microenvironment, provides nutrients to cancer cells, and leads to acidosis, inflammation, angiogenesis, immunosuppression, and radiation resistance.” Include reference.

Line 289: “which may state why diabetic subjects…” Replace with: “which may explain why diabetic subjects…”.

Line 291: “In order for lactic acid and pyruvic acid to react, an enzyme called LDH is required.” Please remove, this explanation is not necessary here.

Line 298: “A major reason for the development of cancer is that the immune system loses its ability to effectively eliminate aberrant cells. High levels of lactate have a deleterious effect on immune cells infiltrating the tumor.” Include reference.

Line 317: “Increasing IL -6 results in an anti-erythropoietic effect, as this cytokine alters the sensitivity of precursor cells to erythropoietin and promotes cellular death of immature erythrocytes, leading to a further decrease in the number of circulating erythrocytes and a decrease in circulating hemoglobin.” Include reference.

Line 332: “In a study…” Replace with: “In a previous study…”.

Line 335: “There was no difference between patients with colorectal cancer who had diabetes and those who did not, according to a recent case-control study on the relationship between type 2 diabetes and colorectal cancer.” What were the parameters evaluated, showing no differences? Also, include the reference.

Line 349: “A high level of glycemia is directly related to the development of an inflammatory state, as evidenced by the increased expression of proinflammatory cytokines such as IL-6, TNFα, and NFκB.” Reference needed.

Line 352: “Studies show that the longer the disease persists and/or the poorer the glycemic control, the more severe the inflammatory process.” Please include references of these studies.

Line 353: “The increase in pro-inflammatory cytokines plays a major role in insulin resistance and leads to the occurrence of diabetic macrovascular cardiovascular and microvascular complications and anemia.” Reference needed.

Line 409, Data Availability Statement: Please include information or eliminate this section.

Line 414, Acknowledgments: Please include information or eliminate this section.

Line 417, Conflicts of Interest: Please include information as requested.

References: Have a thorough look, since there are some references not standardized as the others. For instance, references 23, 24 and 29 have the publication year in bold, others have the year of publication in the wrong place (references 26 and 27). There is also a consistent issue of words that are juxtaposed (which should be separated, for instance in references 1, 8, 10, 12, 16…).

The quality of the English should be improved. I have included some changes/suggestions regarding these issues in the minor comments.

Author Response

Dear reviewer, we are very thankful to you for the pertinent notes; we have carefully read the comments and have revised/ completed the manuscript accordingly. References have been also added, and the other ones have been revised.

Thank you for your valuable major suggestion. A paragraph has been added, in the introduction, evaluating LDH, CEA, and CRP in diabetic patients, without cancer: 

Several studies have suggested the importance of inflammatory biomarkers in the onset of diabetes and the monitoring of anti-diabetic treatment, without necessarily being related to a form of cancer. Thus, elevated C-reactive protein levels have been associated with an increased risk for the development of diabetes [14], and LDH is a biomarker of glycemic variability in diabetes monitoring [15]. There was also a significant increase in CEA values in patients with unbalanced diabetes, with average glycosylated hemoglobin values above 9% [16].

Minor revisions: all your valuable suggestions, step by step, have been corrected/modified, including suggested graphic issues, English language suggestions, and references. 

“There was no difference in survival between patients with colorectal cancer who had diabetes and those who did not, according to a recent case-control study on the relationship between type 2 diabetes and colorectal cancer.” What were the parameters evaluated, showing no differences? Also, include the reference.

Thank you for your observation. The text has been updated, with yellow-marked words added. 

Thank you for taking the time to review our paper and for your valuable suggestions. Overall, we believe that thanks to your pertinent comments, the quality of the manuscript has been improved. For any other questions or suggestions, we are at your disposal.

Cordially yours,

The corresponding authors

Reviewer 2 Report (New Reviewer)

The study provides information on the relationship between diabetes and cancers. A few typing mistakes are found in the manuscript. Other minor concerns are: Two figures 7 are in the manuscript. The average value of H1bc in subgroup 2 is not mentioned in the result. There is no CEA result in the figure. The authors should give the normal range value of hemoglobin to give readers an idea of anemia.

A few typing mistakes are found in the manuscript.

Author Response

Dear reviewer, we would like to thank you for your valuable comments which helped us improve the manuscript.

English mistakes were reconsidered. 

Two figures 7 are in the manuscript. Thank you for your observation. Corrections have been made. 

The average value of H1bc in subgroup 2 is not mentioned in the result. Thank you also, for your valuable comment. A phrase has been added: Non-diabetic patients were considered those with HbA1c values lower than 5.7%.

There is no CEA result in the figure. Thank you for this observation; Figure 9 was added, and this phrase also: Average CEA levels for pancreatic cancer were higher in the diabetic group (39.23 ng/ml vs 12.95 in the control group); In the colon cancer group, diabetic patients had a significantly lower rate of CEA, compared with the control group (44.63 ng/ml vs. 288.85ng/ml).

The authors should give the normal range value of hemoglobin to give readers an idea of anemia. A small commentary was added: The normal range considered was 12.5-14.5 g/dl.

Thank you for taking the time to review our paper and for your valuable suggestions. Overall, we believe that thanks to your pertinent comments, the quality of the manuscript has been improved. For any other questions or suggestions, we are at your disposal.

Cordially yours,

The corresponding authors

Round 2

Reviewer 1 Report (New Reviewer)

I think the authors addressed the issues I pointed, therefore the article should now be considered for publication, from my part.

This manuscript is a resubmission of an earlier submission. The following is a list of the peer review reports and author responses from that submission.

Round 1

Reviewer 1 Report

Introduction section: This needs improvement and authors should focus on describing why this study is important based on previous studies. There is no need to waste time to describe diabetes as a risk factor for cancer as study subjects in group 1 already have both

Study objectives: This is quite confusing as the aim in the introduction section is study " investigate the biological implications of diabetes on the the evolution and prognosis of cancer". None of these aspects was described in the results. At the beginning of the discussion section, authors seem to have changed the study objective to be " to assess the utility of CEA, CRP, albumin level, LDH and hemoglobin as biomarkers of cancer risk". This are markers of inflammation; with diabetes and cancer both being inflammatory conditions. The results are not surprising 

Methods: Authors state that they selected statistically significant sample sample comparing wo groups. There is no justification to indicate that the the sample size was statistically significant

Results: Analysis involving individual groups of cancer seem unjustified/limited given the small numbers being compared

Discussion: It is flawed as authors are discussing some aspects not in their objects. As authors did not analyze any aspects of prognosis. This is also left out i the discussion

Author Response

Dear Academic Editor,

Dear Peer-Reviewers,

We are very thankful to you for the pertinent notes; we have carefully read the comments and have revised/ completed the manuscript accordingly. Our responses are given in a point-by-point manner below, as well, all the changes to the manuscript are highlighted in yellow.

Introduction section: This needs improvement and authors should focus on describing why this study is important based on previous studies. There is no need to waste time describing diabetes as a risk factor for cancer as study subjects in group 1 already have both

Thank you for this observation. Corrections were made for more consciousness. We think that inflammatory biomarkers could be predictive factors for cancer in diabetic patients and can guide the oncologist in terms of the patient's prognosis at the onset of treatment.  We believe that these biomarkers can also be used by a diabetes doctor to be able to suspect the presence of cancer, especially when a glycemic imbalance occurs. 

The study objectives and Methods paragraph were also updated.

Results: Analysis involving individual groups of cancer seem unjustified/limited given the small numbers being compared

Thank you also, for your your valuable comment. 

We choose to make comparisons in the analyzed subgroups because they represent the main types of cancer that are associated with diabetes, and whose subsequent evolution is influenced by metabolic syndrome. The main analysis of the study takes place at the level of the whole group. Also, a supplementary analysis was made, for the treatment period and survival. 

Thank you for taking the time to review our paper and for your valuable suggestions. Overall, we believe that thanks to your pertinent comments, the quality of the manuscript has been improved, and its revised version warrants publication in Diagnostics.

Reviewer 2 Report

• In terms of grammar, the English language of the paper should be improved. • The "Introduction" part of the study should be expanded, considering the research objectives, problems, and hypotheses. • The primary output/endpoint variable(s)/measurement(s) of the study should be defined.  • "The study group was represented by 434 patients" was reported. How was the sample size determined? This information should be explained in the Materials and Methods section.

• Which sampling (probable or non-probable, etc.) method was used in the study? 

• Statistical tests for hypothesis testing and their assumptions should be specified in the study's statistical analysis in the Materials and Methods section. 

• The details (version, license number, etc.) of the statistical package(s) or program(s) should be given in the section of "Data Analysis or Statistical Analysis". 

• It should be explained how the qualitative and quantitative data are summarized under the sub-heading of Statistical Analyses in the Materials and Methods section of the study.  • Data analysis or Statistical analysis sub-section title should be added to the Materials and Methods.  • The exact P values should be added to the table(s) (e.g., p=0.25; p=0.03). • In terms of grammar, the English language of the paper should be improved. • The "Introduction" part of the study should be expanded, considering the research objectives, problems, and hypotheses. • The primary output/endpoint variable(s)/measurement(s) of the study should be defined.  • "The study group was represented by 434 patients" was reported. How was the sample size determined? This information should be explained in the Materials and Methods section.

• Which sampling (probable or non-probable, etc.) method was used in the study? 

• Statistical tests for hypothesis testing and their assumptions should be specified in the study's statistical analysis in the Materials and Methods section. 

• The details (version, license number, etc.) of the statistical package(s) or program(s) should be given in the section of "Data Analysis or Statistical Analysis". 

• It should be explained how the qualitative and quantitative data are summarized under the sub-heading of Statistical Analyses in the Materials and Methods section of the study.  • Data analysis or Statistical analysis sub-section title should be added to the Materials and Methods.  • The exact P values should be added to the table(s) (e.g., p=0.25; p=0.03).

Author Response

Dear Academic Editor,

Dear Peer-Reviewers,

We are very thankful to you for the pertinent notes; we have carefully read the comments and have revised/ completed the manuscript accordingly. Our responses are given in a point-by-point manner below, as well, all the changes to the manuscript are highlighted in yellow.

In terms of grammar, the English language of the paper should be improved.
Thank you for your suggestion. Corrections were made. 

The "Introduction" part of the study should be expanded, considering the research objectives, problems, and hypotheses.
Thank you also, for your your valuable comment. We think that inflammatory biomarkers could be predictive factors for cancer in diabetic patients and can guide the oncologist in terms of the patient's prognosis at the onset of treatment [4]. We believe that these biomarkers can also be used by a diabetes doctor to be able to suspect the presence of cancer, especially when an unexplained glycemic imbalance occurs
Also, in the material and methods section, the group of selected patients was more clearly explained.

 The exact P values should be added to the table(s) (e.g., p=0.25; p=0.03).
Thank you for this observation. More information on the correlations between inflammation biomarkers and data on treatment duration and survival has been added, including graphs containing P-index values. Thank you for taking the time to review our paper and for your valuable suggestions. Overall, we believe that thanks to your pertinent comments, the quality of the manuscript has been improved, and its revised version warrants publication in Diagnostics.
